# Trends and determinants of the triple burden of malnutrition in Ghana; Analyses of two decades of demographic and health survey datasets

Hammond Yaw Addae[1,2]*, Rafatu Tahiru[1,3], Afizu Alhassan[4], Abdul-Ganiyu Fuseini[5], Mohammed Iddrisu[2], Wilhelmina Mensah[6], Fusta Azupogo[7], Martin Nyaaba Adokiya[8]

1 Department of Biochemistry, College of Science, Kwame Nkrumah University of Science and Technology, Kumasi, Ghana, 2 Nursing and Midwifery Training College, Salaga, Ghana, 3 Community Health Nurses Training College, Tamale, Ghana, 4 Faculty of Health Sciences, School of Nursing, Midwifery, and Paramedicine, Curtin University, Perth, Western Australia, Australia, 5 School of Nursing, Midwifery & Social Sciences Central Queensland University, Melbourne, Victoria, Australia, 6 University of Ghana Medical School, University of Ghana, Accra, Ghana, 7 Department of Family and Consumer Sciences, Faculty of Agriculture, University for Development Studies, Tamale, Ghana, 8 Department of Epidemiology, Biostatistics and Disease Control, School of Public Health, University for Development Studies, Tamale, Ghana

* hamondd@yahoo.com

## Abstract

Anaemia, undernutrition and obesity remain complex public health challenges. Their coexistence among households, commonly known as the triple burden of malnutrition (TBM) is a new concept that lacks scholarship within the maternal and child nutrition discourse in Ghana. This study therefore aimed to evaluate the trends and factors associated with TBM among mothers and their children aged 0–59 months in Ghana. This study combined and analysed datasets from the Ghana Demographic and Health Survey from 2003 to 2022. Prevalence and multivariable logistic regression were used to evaluate the trends and determinants of TBM among 11,925 mother-child pairs using complex sample procedures. The pooled prevalence of TBM was 6.7% (5.7 - 6.7). This reduced from 7.6% (6.6 - 8.7) in 2003 to 5.0% (4.1-6.2) in 2022. Male children [AOR 2.23, 95% CI:1.33 - 3.74, p = 0.002] were more likely to suffer TBM than female children. Large birth size [AOR 0.30, 95% CI: 0.17 - 0.54, p < 0.001] and breastfed children [AOR 0.13, 95% CI:0.05 - 0.34, p < 0.001] were less likely to suffer TBM. Women with no education [AOR 5.14, 95% CI:1.16 - 22.75, p < 0.031] and those with inadequate dietary diversity [AOR 2.53, 95% CI:1.50 - 4.26, p < 0.001] were more likely to suffer TBM. Also, high-wealth [AOR 0.13, 95% CI:0.05 - 0.33, p < 0.001] and rural households [AOR 0.34, 95% CI: 0.05 - 0.33, p < 0.001] were less likely to suffer TBM. Although the prevalence of TBM reduced over the past two decades, the pooled estimate remains high in Ghana. The associated factors include breastfeeding, childbirth size, maternal education, dietary diversity, wealth

**Data availability statement:** The data for this study are publicly available at https://dhsprogram.com/data/available-datasets.cfm. Interested persons can reproduce our study's findings in its entirety by obtaining these datasets from original owners directly and replicating our methods section. The free access granted authors are applicable to all other interested researchers, on request from the original owners of the datasets.

**Funding:** The author(s) received no specific funding for this work.

**Competing interests:** The authors have declared that no competing interests exist.

and urbanicity. Strategies that promote breastfeeding, improve dietary diversity and ensure equitable distribution of resources are urgently needed to mitigate the TBM.

## Introduction

Malnutrition remains a complex public health challenge globally. Malnutrition mainly manifests as micronutrient deficiencies, undernutrition, overweight or obesity primarily among women and children [1]. Currently, as acknowledged by the United Nations Children's Fund (UNICEF) [2], the world has to contend with a peculiar nutrition challenge in which individuals are experiencing a concurrent existence of all three forms of malnutrition in the same household. This is commonly known as the triple burden of malnutrition (TBM) [3,4], a systemic consequence of shared underlying factors emanating from food systems inadequacies and socioeconomic disparities.

Although a fairly new concept, TBM is gaining traction as a public health concern, with stagnating undernutrition rates, high iron deficiency anaemia rates and significant increases in overnutrition occurring concurrently [5]. At the global level for instance, one in five children under five years was estimated to be stunted as of 2021, with two in five children and one in three women of reproductive age experiencing anaemia [5,6]. Paradoxically, 6% of the world's population under five years are overweight and this is projected to reach a staggering 51% by 2035 [5,7].

The global prevalence of TBM is imprecise, it is however estimated to be about 12% among low-and-middle-countries (LMICs) and 25% among some Sub-Saharan African (SSA) countries [8,9]. These rates are notably high particularly when juxtaposed against the provisions of the Sustainable Development Goal 2.2 which enjoins countries to end all forms of malnutrition by 2030 [10]. More importantly, LMICs have had the slowest reductions in undernutrition and micronutrient deficiencies and remain vulnerable to overnutrition and their concurrent existence [11].

Malnutrition in all forms poses significant health risks across the lifespan, affecting individuals, families, and the entire population. For instance, undernutrition during childhood has been linked to a higher likelihood of impaired cognitive development, while childhood excess weight is known to increase the risks of chronic non-communicable diseases later in life [12,13]. Similarly, maternal micronutrient deficiencies and obesity are associated with negative health outcomes for both mother and foetus [14,15]. As a result, the existence of maternal micronutrient deficiencies and obesity in an individual could burden health systems and retard progress in achieving nations' set health and developmental targets.

While LMICs are more likely to be disproportionately affected by the TBM due to the pronounced effects of a rapid nutrition transition, there is paucity of literature on the factors that drive TBM at the household, maternal and child-level in Ghana. Despite growing recognition of TBM as a public health challenge, research on its long-term trends and underlying factors remains limited. One major gap in the literature is the limited depth of analysis on TBM, with most studies providing only a superficial snapshot without adequately examining medium-to-long-term trends and their

cumulative associated factors. For instance, a study analysing a decade of Demographic and Health Surveys (DHS) data from 32 SSA countries, Ahinkorah et al., (2021) [9] identified child-related factors such as age, sex, and birth size, as well as maternal factors such as education level, antenatal attendance, and household cooking fuel, as significant influencers of TBM. However, Ahinkorah et al.'s study was not country-specific enough, making it difficult to apply the conclusions to Ghana. The lack of country-specific data on TBM has made it even more difficult to implement public health approaches aimed at mitigating the consequences of TBM. As a comparatively new concept, with limited scholarship within previous maternal and child malnutrition discourse in Ghana, it is important to ascertain the trend of TBM over the past two decades.

To address this gap, the present study aimed to evaluate trends in malnutrition and identify the factors associated with the pooled estimate of TBM in Ghana using the Ghana DHS 2003, 2008, 2014 and 2022 datasets. By providing a more comprehensive analysis, this study seeks to inform sustainable solutions for addressing the co-existence of malnutrition, advance scholarly discussions on TBM, and contribute to achieving Sustainable Development Goal 2.2 in Ghana.

## Methods

### Ethics statement

The ICF and the Ghana Statistical Service submitted all the survey protocols to the Ethical Review Committee of the Ghana Health Service, ensuring compliance with ethical research standards in Ghana. Additionally, ICF had previously submitted all DHS protocols to the ICF Institutional Review Board for approval, ensuring adherence to both United States and international ethical research standards. The Institutional Review Board and the Ethical Review Committee approved the ethical clearance for all DHS surveys. Hence, this present study did not require any ethical approval(s) since it is a secondary analysis of these DHS datasets. The datasets granted to authors was anonymized such that the authors did not have access to data that could identify the individual participants. Additional information on the DHS usage and ethical standards can be accessed online https://dhsprogram.com/methodology/protecting-the-privacy-of-dhs-survey-respondents.cfm [16].

### Study design

The DHS is a nationally representative cross-sectional household survey conducted approximately every 5 years on relevant population, health, nutrition and other indicators among men (15–59 years), women (15–49 years) and children in about 90 LMICs [17,18]. The DHS provides sound data that aids in formulating and monitoring policy and evaluating existing health-related programs' impact. Some previous datasets did not include a measure of anaemia, as such this present study is a secondary analysis of the four most recent DHS conducted in Ghana, i.e., the Ghana DHS 2003 [19], the Ghana DHS 2008 [20], the Ghana DHS 2014 [21] and the Ghana DHS 2022 [22] only. Authorization to these survey datasets was granted upon request from the ICF DHS program on 12th February 2024. These datasets can be accessed for free online at https://dhsprogram.com/data/available-datasets.cfm [23].

### Sampling procedure and sample size

The Ghana DHS 2022 used the updated sampling frame from the Ghana Statistical Service 2021 population and housing census and a two-stage sampling method [22]. This comprised the selection of 618 clusters from the sampling frame using probability proportional to size for rural and urban zones. In the second stage sampling, 10–30 prelisted households were selected using systematic random sampling in each cluster resulting in a maximum of 18,540 households across all 16 regions of Ghana. The allocation of the number of households to be selected in each cluster was also based on probability proportional to size. The number of eligible individuals in all households was 22,580, of whom 22,058 were interviewed in the Ghana DHS 2022. The specific details of the Ghana DHS 2022 sampling methodology have been

extensively described elsewhere [22]. These same two-stage sample procedures were used in the Ghana DHS 2014, 2008 and 2003 and their detailed description have also been published elsewhere [19,20,24]. The total number of house-holds ever included in all Ghana DHS is 66,419 with 74,186 respondents interviewed. Due to the lack of data on haemo-globin (Hb) levels, the Ghana DHS 1988, 1993 and 1998 datasets with a total of 17,824 households were excluded from this study. Men aged 15–59 years and women in households without children aged 0–59 months were also excluded leaving 12,888 mother-child pairs in their respective households. From 12,888 mother child-pairs, 963 pairs were removed due to missing data on anthropometry and Hb values. Such exclusions were carried out under the assumption that these values were 'missing completely at random' as stipulated by Mack et al., [25]. In all, 11,925 households with mother-child pairs were retained for analysis. The detailed selection process flow chart is presented in Fig 1.

## Data collection

All included Ghana DHS datasets used an interviewer-administered computer-assisted procedure to administer pre-tested structured questionnaires to eligible respondents. According to the published reports of the Ghana DHS, four main questionnaires were used to collect data from respondents; the household questionnaire, the man's questionnaire, the woman's questionnaire and the biomarker questionnaire [22]. These questionnaires were further divided into sections. Specifically, this study was interested in data collected from the household registry section of the household questionnaire. It also was interested in aspects of the women's questionnaire in which data were collected on reproduction, child health and nutrition, pregnancy, postnatal care, and other health-related factors. In the 2022 report, anthropometry, i.e., weight and height for eligible women and children 0–59 months were measured using the SECA 874U Doctor's scale and the ShorrBoard® measuring board, respectively. For anaemia testing, capillary blood samples were collected into microcuvette from mothers 15–49 years who consented as well as children 6–59 months whose guardians had given consent for the testing. The battery-operated Hemocue® 201+testing kit was used onsite to analyse the blood samples taken. Specific details of the data collection processes of the variables included in this study are published elsewhere [22].

## Study variables

This study's variables were selected from the Ghana DHS datasets based on previous publications on TBM [8,9,26–31]. These were broadly categorized into child-level, maternal-level and household-level variables.

## Dependent variables

The main dependent variable was the triple burden of malnutrition. The secondary dependent variables included stunt-ing, wasting, underweight, overweight and anaemia status for children, and underweight, overweight and anaemia status for mothers. In this study, a child is termed undernourished if they are stunted (height-for-age z-score<-2SD), wasted (weight-for-age z-score<-2SD) or underweight (weight-for-height z-score<-2SD) and overweight if weight-for-height z-score>+2SD [32]. Underweight and overweight among women were defined as Body Mass Index (BMI) < 18.5 kg/m$^2$ and BMI ≥ 25 kg/m$^2$, respectively [33]. Additionally, anaemia among children 6 – 59 months and women 15 – 49 years were both defined by the Hb concentration where those with Hb < 11.0 g/dL and Hb < 12.0 g/dL were termed anaemic, respectively [6]. For household burdens of malnutrition, household undernutrition is defined as when a household has an undernourished child or an underweight woman. And household overnutrition occurs when the household has an over-weight child or an overweight woman. Meanwhile, anaemic households are households with either an anaemic child or an anaemic woman of reproductive age. Households with the co-existence of household undernutrition, overnutrition and anaemia are then defined as households with TBM [26]. For instance, a household is said to have TBM if an overweight woman and an anaemic child who is stunted are in the same household. Likewise, an overweight child and a woman who is underweight and anaemic in the same household could also be classified as a household with TBM.

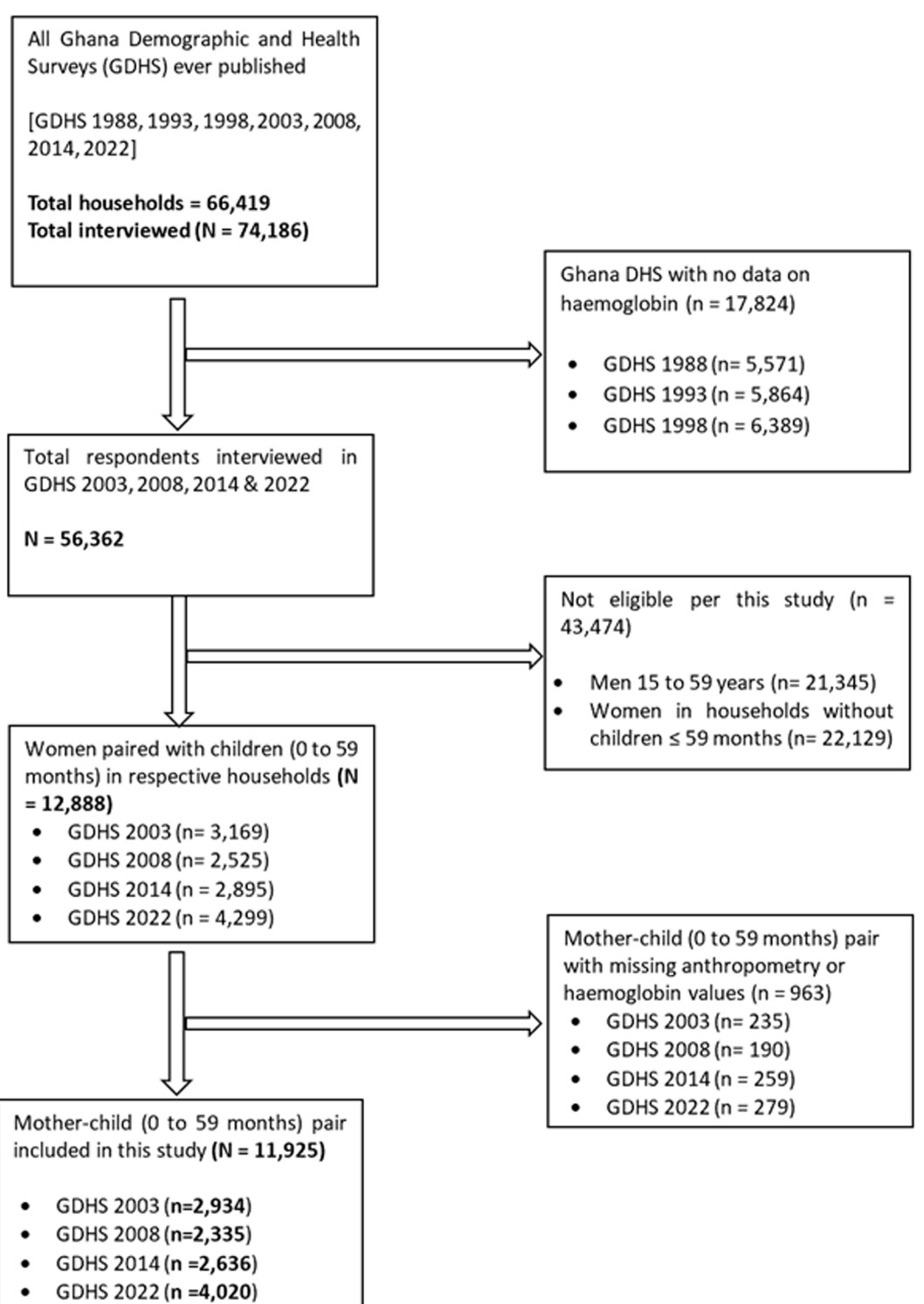

**Fig 1. Flow chart showing selection of mothers and children aged 0 to 59 months pair in Ghana Demographic and Health Survey 2003, 2008, 2014 & 2022 datasets.**

## Independent variables

**Child-level variables.** These included child's sex, child age in months, child's estimated birth size as recalled by women (described as large, average or small), child's birth order and child's breastfeeding status grouped as still breastfeeding, ever breastfed or never breastfed. Child's dietary diversity was also included, in which dietary diversity was measured to reflect the quality of food intake using the 24-hour dietary recall method. For Ghana DHS 2003, 2008 and 2014 datasets, the seven-food group WHO indicator was used as a measure of dietary diversity [34]; 1) grains, starchy roots, tubers and plantains 2) beans, nuts and seeds 3) dairy products 4) meat, fish and poultry 5) eggs 6) vitamin-A rich fruits and vegetables 7) other fruits and vegetables. With this, children who consumed food items from ≥ 4 food groups were deemed to have adequate dietary diversity. For the Ghana DHS 2022 dataset, the recent eight-food group WHO indicator [35] was used, and children who consumed ≥ 5 food groups were classified as having adequate dietary diversity. Compared to the eight-food group indicator, the seven-food group indicator did not include breastmilk as a food group. These indicators have been validated within low- and middle-income country context as a tool that reflects access and overall quality of food intake among children 6–23 months.

**Maternal level variables.** The variables assessed under maternal factors included the age of women in years, categorized as < 25, 25 – 34 and ≥35. The marital status (dichotomized), maternal education (no education, primary, secondary and tertiary), employment (dichotomized), breastfeeding (dichotomized) and whether the last delivery was a caesarean section or not were included. This study also included the dietary diversity score of women which was assessed using the validated women's dietary diversity tool [36]. This was calculated based on 24 hr-food consumption patterns from the FAO & FHI 10-food group classifications [36]; 1) grains, white roots and tubers, and plantains, 2) beans, peas and lentils, 3) nuts and seeds 4) milk and milk products 5) meat, poultry and fish, 6) eggs, 7) dark green leafy vegetables 8) other vitamin A rich fruits and vegetables, 9) other vegetables and 10) other fruits. The summed scores ranged in ascending order of magnitude from 0 to 10, where they were scored one for each food group consumed, or else zero. Women who consumed a score of ≥ 5 food groups were deemed to have adequate dietary diversity and < 5, inadequate dietary diversity.

**Household-level variables.** The following household factors were also assessed; number of household members, age and sex of household head, household wealth (categorized as low, middle or high), urbanicity (either urban household or rural household) and region of household, i.e., the previous 10 regions of Ghana. Where the regions in the Ghana DHS 2022 dataset were regrouped from 16 to 10 regions to conform with previous datasets. The households were further categorized into climatic zones based on the regions they are situated. Households located in Northern, Upper East and Upper West were categorized as savannah climatic zone, those in Eastern, Ashanti and Brong Ahafo regions were termed Forest climatic zone and households in Western, Central, Greater Accra and Volta regions were defined as coastal climatic zone in tandem with literature [37,38]. According to the Ghana DHS, the household wealth index was derived by the principal component analysis method, where household assets were used as a basis to determine wealth [22]. The sources of drinking water were categorized as protected water sources and unprotected water sources. And sanitation was classified as improved facilities, unimproved toilet facilities or open defecation. These water and sanitation variables were assessed and classified based on WHO and UNICEF recommendations [39].

## Statistical analysis

All statistical analyses were conducted using the SPSS version 22 (SSPS Inc. Chicago, IL, USA). First, the relevant variables in each of the four datasets were merged into one combined dataset such that each variable is a continuation of itself from 2003 to 2022. This was done manually by copying each relevant variable from all datasets into the combined dataset. In the combined dataset, all variables in each year's dataset were designated as such by the year of data collection. Descriptive statistics were then used to summarize the participant's background characteristics, where categorical

variables were presented as frequencies and percentages, and continuous variables as mean and standard errors. The trend of all forms of malnutrition including the TBM were presented as frequencies and percentages in their corresponding years of data collection. The pooled prevalence of each type of malnutrition was calculated by summing up the estimates from all four points of data collection. Bivariate binary logistic regression analyses were used to determine the unadjusted odds of all the malnutrition indicators in 2022, 2014 and 2008 relative to 2003. Additional bivariate logistic regression analyses were conducted between the main dependent variable and the independent variables to identify the significant variables for inclusion in the multivariable model. Child-level, maternal-level and household-level factors that were significant with $p < 0.05$ in the bivariate analyses were included in the multivariable logistic regressions all at once. The model was assessed for multicollinearity using a correlation matrix. Our analyses accounted for the multistage cluster design of the DHS by adjusting for clustering and stratification using the SPSS Complex Samples module. This procedure involved creating complex sample plan file (csplan) based on cluster, strata and sample weight variables in the combined dataset. The csplan file was used as the basis for all analyses in this study. The purpose was to normalize variance and mitigate the risk of bias in under and over sampled subpopulations. All analyses were conducted with a 95% Confidence Interval (CI), with statistical significance set at p-value $< 0.05$ for the bivariate and multivariable analyses. The chi-square value and overall model p-value of the Wald test were used to assess model significance.

## Results

### Background characteristics

Table 1 shows 11,925 mother-child pairs in the same household from 2003, 2008, 2014 and 2022 Ghana DHS datasets. The mean age of children and women were $28.37 \pm 0.17$ months and $30.79 \pm 0.08$ years, respectively. The proportion of males and females were similar with prevalence of 49.8% and 50.2%, respectively. Also, 74.7% of children aged $< 24$ months were still breastfeeding, 22.5% had ever breastfed and 2.8% had never been breastfed. For education, about half of the mothers had at least a secondary school education while 30.8% had no education. About 4 in 10 mothers (42.0%) had adequate dietary diversity and about 1 in 10 (10.3%) reported having given birth through caesarean delivery. The mean household size was $5.78 \pm 0.04$. About 8 in 10 (79.5%) households had protected source of main drinking water while 20.5% had unprotected source of drinking water. For toilet facilities, 50.5% had improved toilet facilities, 21.5% had unimproved toilet facilities and 28.1% practised open defecation. Additionally, 24.3%, 59.2% and 16.5% classified as low, middle and high wealth, respectively. Approximately, 55% of respondents lived in urban areas and 4 in 10 in coastal climatic zone.

### Prevalence and trends of the burden of malnutrition in Ghana from 2003 to 2022

The pooled estimate of stunting among children was 23.3% (22.3 - 24.3), having reduced from 34.5% (32.7 - 36.4) in 2003 to 16.4% (14.9 - 18.1) in 2022 (Table 2). Anaemia among children also reduced from 72.0% (70.0 - 74.0) in 2003 to 46.2% (43.9 - 48.6) in 2022 with a pooled prevalence of 61.7% (60.7 - 71.7). Similar downward trends can be observed for child wasting, underweight and overweight with pooled prevalences of 6.8% (6.3 - 7.3), 13.1% (12.1 - 12.1) and 3.4% (2.9 - 4.9), respectively. For mothers, a reduction is observed in underweight, i.e., from 8.4% (7.1 - 9.8) in 2003 to 5.0% (4.1 - 6.0) in 2022 with a pooled prevalence of 6.8% (5.8 - 5.8). A reduction is also observed in anaemia, i.e., from 45.7% (43.2 - 48.2) in 2003 to 40.2% (37.9 - 42.5) in 2022, with a pooled prevalence of 46.4% (44.4 - 57.4). However, mothers overweight almost doubled from 25.1% (22.7 - 27.6) in 2003 to 45.0% (42.2 - 47.8) in 2022. Mothers overweight had a pooled prevalence of 35.2% (34.2 - 26.2).

Household undernutrition and household anaemia had pooled prevalences of 33.4% (31.4 - 36.4) and 74.8% (73.8 - 82.8), respectively, with both witnessing reductions from 2003 to 2022. However, household overweight increased from 27.8% (25.4 - 30.3) in 2003 to 45.8% (43.1 - 48.6) in 2022 with a pooled prevalence of 37.2% (36.3 - 29.3). TBM was

**Table 1. Background characteristics of Ghana Demographic and Health Survey 2003, 2008, 2014 and 2022 (weighted n = 11,925).**

| Variable | Frequency (%) or mean ± Standard Error |
|---|---|
| **Child-level variables** | |
| **Mean child's age (months)** | 28.37 ± 0.17 |
| **Child's age (months)** | |
| 0–5 | 1329 (11.1) |
| 6–11 | 1175 (9.9) |
| 12–23 | 2626 (22.0) |
| 24–59 | 6795 (57.0) |
| **Child's sex** | |
| Male | 5936 (49.8) |
| Female | 5989 (50.2) |
| **Child's birth size (recall)** | |
| Large | 4959 (41.6) |
| Average | 3883 (32.5) |
| Small | 1548 (13.0) |
| No response | 1535 (12.9) |
| **Mean birth order** | 2.85 ± 0.04 |
| **Mean child dietary diversity** | 2.70 ± 0.04 |
| **Child's dietary adequacy (6–23 months) (n = 3801)\*** | |
| Inadequate | 2552 (67.1) |
| Adequate | 1249 (32.9) |
| **Child breastfeeding status (0–23months) (n = 5,130)** | |
| Still breastfeeding | 3832 (74.7) |
| Ever breastfed | 1156 (22.5) |
| Never breastfed | 142 (2.8) |
| **Maternal-level characteristics** | |
| **Mean women age (years)** | 30.79 ± 0.08 |
| **Women age (years)** | |
| <25 | 2382 (20.0) |
| 25–34 | 5888 (49.4) |
| 35+ | 3655 (30.6) |
| **Marital status** | |
| Married | 9651 (80.9) |
| Not married | 2274 (19.1) |
| **Education** | |
| No education | 3669 (30.8) |
| Primary | 2352 (19.7) |
| Secondary | 5364 (45.0) |
| Higher | 540 (4.5) |
| **Employed** | |
| No | 1969 (16.5) |
| Yes | 9956 (83.5) |
| **Currently breastfeeding** | |
| No | 8092 (67.9) |
| Yes | 3833 (32.1) |
| **Mean women dietary diversity** | 3.34 ± 0.04 |
| **Women-minimum dietary diversity** | |
| Inadequate (< 5 groups) | 6914 (58.0) |
| Adequate (≥ 5 groups) | 5011 (42.0) |

*(Continued)*

| Variable | Frequency (%) or mean±Standard Error |
|---|---|
| **Last birth caesarean delivery** | |
| No | 9859 (82.7) |
| Yes | 1236 (10.3) |
| No response | 831 (7.0) |
| **Household-level characteristics** | |
| **Mean household size** | 5.78±0.04 |
| **Mean age of household head (years)** | 40.39±0.18 |
| **Sex of household head** | |
| Male | 8595 (72.0) |
| Female | 3329 (28.0) |
| **Main source of drinking water** | |
| Protected | 9479 (79.5) |
| Unprotected | 2446 (20.5) |
| **Type of toilet** | |
| Open defecation | 3350 (28.0) |
| Unimproved | 2558 (21.5) |
| Improved | 6017 (50.5) |
| **Household wealth** | |
| Low | 2897 (24.3) |
| Middle | 7058 (59.2) |
| High | 1970 (16.5) |
| **Urbanicity** | |
| Urban | 6521 (54.7) |
| Rural | 5404 (45.3) |
| **Region of household∞** | |
| Western | 1144(9.6) |
| Central | 1177(9.9) |
| Greater Accra | 1500(12.6) |
| Volta | 929(7.8) |
| Eastern | 1044(8.7) |
| Ashanti | 2184(18.3) |
| Brong Ahafo | 1238(10.4) |
| Northern Region | 1784(14.9) |
| Upper East | 511(4.3) |
| Upper West | 414(3.5) |
| **Climatic Zones** | |
| Savannah | 2708 (22.7) |
| Forest | 4466 (37.5) |
| Coastal | 4750 (39.8) |
| **Year of survey** | |
| 2003 | 2934 (24.6) |
| 2008 | 2335 (19.6) |
| 2014 | 2636 (22.1) |
| 2022 | 4020 (33.7) |

*Adequacy was≥4 food groups for 2003, 2008 and 2014 datasets and ≥ 5 food groups for 2022 dataset; it was restricted to children aged 6–23months.

∞The regions in the 2022 dataset have been regrouped from 16 to the 10 regions to conform with previous datasets.

**Table 2. Prevalence of the burden of malnutrition in Ghana from 2003, 2008, 2014 and 2022 datasets.**

| Burden of malnutrition | Year 2003 (n=2,934) | | Year 2008 (n=2,335) | | Year 2014 (n=2,636) | | Year 2022 (n=4020) | | Pooled estimate (N=11,925) | |
|---|---|---|---|---|---|---|---|---|---|---|
| | n | % (95% CI) | n | % (95% CI) | n | % (95% CI) | n | % (95% CI) | n | % (95% CI) |
| **Child under 5 years** | | | | | | | | | | |
| A. Stunted (HAZ<-2SD) | 1014 | 34.5% (32.7-36.4) | 634 | 27.2% (24.8-29.7) | 470 | 17.8% (15.9-19.9) | 660 | 16.4% (14.9-18.1) | 2778 | 23.3% (22.3-24.3) |
| B. Wasted (WHZ<-2SD) | 240 | 8.2% (7.2-9.3) | 201 | 8.6% (7.5- 9.9) | 122 | 4.6% (3.6-5.9) | 246 | 6.1% (5.2-7.2) | 809 | 6.8% (6.3-7.3) |
| C. Underweight (WAZ<-2SD) | 515 | 17.5% (15.9-19.3) | 318 | 13.6% (12.0- 15.3) | 284 | 10.8% (9.1-2.5) | 483 | 12.0% (10.8-13.4) | 1599 | 13.1% (12.1-12.1) |
| D. Overweight (WHZ>+2SD) | 118 | 4.0% (3.3-4.9) | 118 | 5.1% (4.0- 6.4) | 72 | 2.7% (2.0-3.7) | 78 | 1.9% (1.5-2.5) | 386 | 3.4% (2.9-4.9) |
| E. Anaemic (Hb<11.0g/dl) (5 to 59m) | 2080 | 72.0% (70.0-74.0) | 1572 | 74.1% (71.6-76.4) | 1535 | 62.2% (59.0-65.4) | 1757 | 46.2% (43.9-48.6) | 6944 | 61.7% (60.7-71.7) |
| **Woman (15–49 years)** | | | | | | | | | | |
| F. Underweight (BMI<18.5kg/m$^2$) | 246 | 8.4% (7.1 - 9.8) | 163 | 7.0% (5.8- 8.3) | 118 | 4.5% (3.6-5.6) | 200 | 5.0% (4.1-6.0) | 727 | 6.8% (5.8 - 5.8) |
| G. Overweight (BMI≥25.0kg/m$^2$) | 736 | 25.1% (22.7-27.6) | 672 | 28.8% (26.0- 1.7) | 1047 | 39.7% (36.2-43.3) | 1809 | 45.0% (42.2-47.8) | 4264 | 35.2% (34.2-26.2 |
| H. Anaemic (Hb<12.0g/dl) | 1341 | 45.7% (43.2-48.2) | 1410 | 60.4% (57.5-63.2) | 1139 | 43.2% (40.2-46.3) | 1614 | 40.2% (37.9-42.5) | 5503 | 46.4% (44.4-57.4) |
| **Household** | | | | | | | | | | |
| I. Household undernutrition (A or B or C or F) | 1326 | 45.2% (43.0- 47.4) | 903 | 38.7% (36.1-41.3) | 690 | 26.2% (23.9-28.6) | 1025 | 25.5% (23.6-27.4) | 3944 | 33.4% (31.4-36.4) |
| J. Household overnutrition (D or G) | 815 | 27.8% (25.4- 30.3) | 745 | 31.9% (29.1-34.9) | 1086 | 41.2% (37.8-44.7) | 1842 | 45.8% (43.1-48.6) | 4489 | 37.2% (36.3-38.1) |
| K. Household anaemia (E or H) | 2393 | 81.6% (79.8- 83.2) | 1976 | 84.6% (82.7-86.4) | 1942 | 73.7% (70.9-76.3) | 2563 | 63.8% (61.5-66.0) | 8874 | 74.8% (73.8-82.8) |
| **Triple burden of malnutrition (I and J and K)** | 223 | 7.6% (6.6- 8.7) | 180 | 7.7% (6.5- 9.2) | 127 | 4.8% (3.9-5.9) | 202 | 5.0% (4.1-6.2) | 732 | 6.7% (5.7-6.7) |

HAZ, height-for-age z score; WHZ, weight-for-height z score; WAZ; weight-for-age z score; BMI, body mass index; Hb, haemoglobin.fv.

7.6% (6.6 - 8.7) in 2003, increased to 7.7% (6.5 - 9.2) in 2008 and reduced to 4.8% (3.9 - 5.9) in 2014. The prevalence as of 2022 was 5.0% (4.1 - 6.2) with a pooled prevalence of 6.7% (5.7 - 6.7).

### The trends of household burdens of malnutrition and triple burden of malnutrition in Ghana in 2003, 2008, 2014 and 2022

Household undernutrition (A) and household anaemia (C) showed a downward trend, while household overnutrition (B) was on an upward trend. The triple burden of malnutrition (D) showed a stagnated trend among mother-child pairs (Fig 2).

### The odds of the burden of malnutrition in 2008, 2014 and 2022 relative to 2003 in Ghana

We conducted binary bivariate logistic regression to determine the likelihood of suffering from some type of malnutrition in 2003, compared to 2008, 2014, and 2022. The results showed that children were less likely to be stunted, wasted, underweight and anaemic in 2022 as compared to 2003. Mothers were also less likely to be underweight and anaemic in 2022 as compared to 2003. However, mothers in 2022 were about 2.5 times more likely to be overweight [OR 2.45, 95% CI: 2.06 - 2.91, p<0.001] as compared to their peers in 2003. For the household level, households in 2022 were 58% less

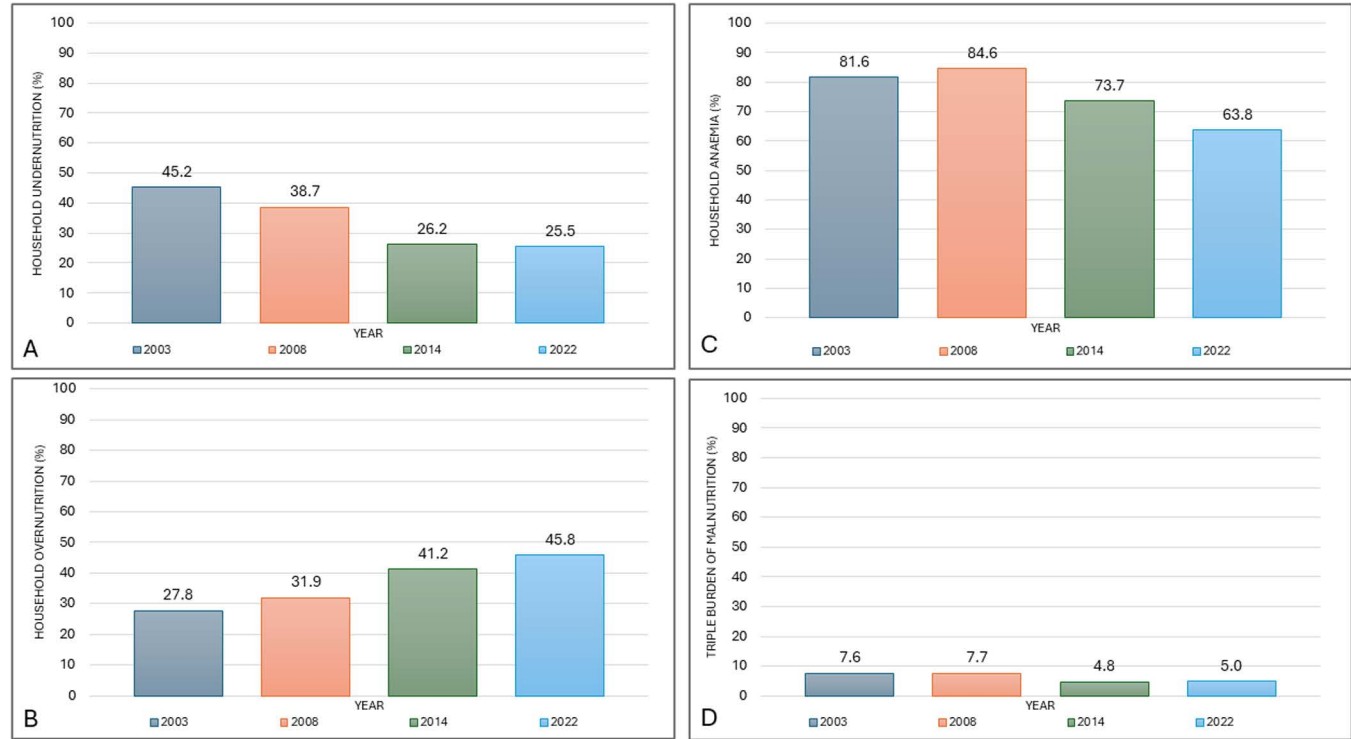

**Fig 2. Trends in the burden of malnutrition among mother-child pairs between 2003-2022.**

likely to be undernourished [OR 0.42, 95% CI: 0.37 to 0.47, p<0.001] and 60% less likely to be anaemic [OR 0.40, 95% CI: 0.34 to 0.46, p<0.001] in 2022 as compared to households in 2003. However, households were 2.2 times more likely to be overweight [OR 2.20, 95% CI: 1.86 to 2.60, p<0.001] in 2022 as compared to those in 2003. Furthermore, households were 36% less likely to suffer TBM in 2022 [OR 0.64, 95% CI: 0.49 to 0.83, p<0.001] and 39% less likely to suffer TBM in 2014 [OR 0.61, 95% CI: 0.47 to 0.80, p<0.001] as compared to households in 2003 (Table 3).

### Determinants of the triple burden of malnutrition in Ghana (2003–2022)

The model showing the likelihood of mother-child pair in a household having TBM and the associated factors are presented in Table 4. For the child-level variables, male children were approximately 2.2 times [AOR 2.23, 95% CI: 1.33 to 3.74, p=0.002] more likely to suffer TBM as compared to their female counterparts. Children with large and average birth sizes were 70% [AOR 0.30, 95% CI: 0.17 to 0.54, p<0.001] and 64% [AOR 0.36, 95% CI: 0.20 to 0.66, p=0.001] less likely to suffer TBM, respectively, as compared to children with small birth size. Children who were still breastfeeding and children who have ever breastfed were 87% [AOR 0.13, 95% CI: 0.05 - 0.34, p<0.001] and 81% [AOR 0.19, 95% CI: 0.09 to 0.39, p<0.001] less likely to suffer TBM, respectively, as compared to never breastfed children. Regarding maternal-level factors, mothers with no formal education and those with only primary education were significantly more likely to experience TBM compared to their counterparts with tertiary education. Specifically, mothers with no education had 5.1 times higher odds [AOR=5.14; 95% CI: 1.16–22.75; p<0.031], while those with primary education had 4.9 times higher odds [AOR=4.88; 95% CI: 1.16–17.58; p<0.015] of experiencing TBM. Mothers with inadequate dietary diversity were 2.5 times [AOR 2.53, 95% CI: 1.50 to 4.26, p<0.001] more likely to suffer TBM as compared with mothers with adequate dietary diversity. At the household level, high-wealth households were 87% less likely [AOR 0.13, 95% CI: 0.05 to 0.33,

**Table 3. The odds of the burden of malnutrition in Ghana Demographic and Health Survey 2008, 2014 and 2022 relative to 2003.**

| Burden of malnutrition | 2003 | 2008 | | 2014 | | 2022 | |
|---|---|---|---|---|---|---|---|
| | | OR (95% CI) | P-value | OR (95% CI) | P-value | OR (95% CI) | P-value |
| **Child under 5 years** | | | | | | | |
| Stunted | Ref. | 0.71(0.61-0.82) | **<0.001** | 0.41 (0.35-0.48) | **<0.001** | 0.37 (0.32 - 0.43) | **<0.001** |
| Wasted | Ref. | 1.06 (0.86-0.30) | 0.575 | 0.55 (0.41-0.73) | **<0.001** | 0.73 (0.59 - 0.92) | **0.006** |
| Underweight | Ref. | 0.74 (0.61-0.89) | **0.001** | 0.57 (0.46-0.69) | **0.001** | 0.64 (0.54 - 0.76) | **<0.001** |
| Overweight | Ref. | 1.27 (0.92-1.75) | 0.148 | 0.67 (0.46-0.96) | **0.029** | 0.47 (0.33 - 0.66) | **<0.001** |
| Anaemia (5–59months) | Ref. | 1.11 (0.95-1.30) | 0.197 | 0.64 (0.54- 0.76) | **<0.001** | 0.33 (0.29 - 0.38) | **<0.001** |
| **Woman (15–49 years)** | | | | | | | |
| Underweight | Ref. | 0.82 (0.63-1.06) | 0.132 | 0.51 (0.38-0.69) | **<0.001** | 0.57 (0.44 - 0.75) | **<0.001** |
| Overweight | Ref. | 1.21 (0.99-1.46) | 0.051 | 1.97 (1.6-2.40) | **<0.001** | 2.45 (2.06 - 2.91) | **<0.001** |
| Anaemia | Ref. | 1.81 (1.55-2.12) | **<0.001** | 0.90 (0.77-1.06) | 0.216 | 0.80 (0.69 - 0.92) | **0.002** |
| **Household** | | | | | | | |
| Household undernutrition | Ref. | 0.76 (0.66-0.87) | **<0.001** | 0.43 (0.37-0.50) | **<0.001** | 0.42 (0.37 - 0.47) | **<0.001** |
| Household overnutrition | Ref. | 1.22 (1.02-1.46) | **0.031** | 1.82 (1.51- 2.20) | **<0.001** | 2.20 (1.86 - 2.60) | **<0.001** |
| Household anaemia | Ref. | 1.24 (1.03-1.49) | **0.019** | 0.63 (0.53- 0.76) | **<0.001** | 0.40 (0.34 - 0.46) | **<0.001** |
| Triple burden of malnutrition | Ref. | 1.02 (0.80 -1.29) | 0.903 | 0.61 (0.47 - 0.80) | **<0.001** | 0.64 (0.49 - 0.83) | **0.001** |

p<0.001] to have TBM as compared to low-wealth households. Additionally, households considered rural were 66% less likely [AOR 0.34, 95% CI: 0.05 - 0.33, p<0.001] to have TBM relative to urban households.

## Discussion

The study analysed trends in the TBM in Ghana using Ghana DHS data from 2003 to 2022 and evaluated the key associated factors. While childhood undernutrition (i.e., stunting, wasting, underweight, overweight) and anaemia (i.e., child and maternal) declined over time, maternal overweight increased significantly. The pooled TBM prevalence was 6.7%, showing an overall decline from 7.6% in 2003 to 5.0% in 2022. Consistent with a previous study among SSA countries [9], factors such as child sex, birth size, breastfeeding, maternal education, dietary diversity, wealth, and urbanicity were significantly associated with TBM in the present study.

To the best of our knowledge, this was the first study to analyse the trends and factors associated with the pooled estimate of the TBM in Ghana. As such direct country-specific literature comparisons may not be feasible. Nonetheless, the pooled TBM estimate of 6.7% in this study aligns with the reported TBM prevalence in Ghana as reported in a multinational study that analysed 23 SSA countries between 2008 and 2017 [26]. Similarly, this finding is consistent with a study conducted in Nepal, where TBM prevalence was 7.0% [27]. However, the TBM prevalence in the present study is lower than that reported in a multi-national study among some LMICs including Indonesia [8,40], but higher than findings from India, and other African countries such as Malawi and Ethiopia [9,28,30,31]. Such contrasts in prevalence could be due to differences in maternal obesity rates. For instance, UNICEF describes Indonesia as having severe TBM owing to the rapid increases in maternal obesity rates which is estimated to be about 45% [41]. In contrast, overweight and obesity rates in India and Malawi are lower at 25% and 19%, respectively [28,42].

TBM has been on a downward trend in Ghana from 7.6% in 2003 to 4.8% in 2014 and remained steady at 5.0% in 2022. In further analysis, TBM had a significantly lower likelihood of occurring in 2022 as compared to 2003. This inconsistency could be an indication of a lack of holistic progress in the reduction of all forms of malnutrition. The reductions in undernutrition and increases in maternal and household overweight have been consistent over the past two decades and this is comparable with the findings of the WHO global report on malnutrition among LMICs [5]. The specific reasons

**Table 4. Multivariable analysis of determinants of the triple burden of malnutrition in Ghana from 2003 to 2022.**

| Variables | AOR | P value |
|---|---|---|
| **Child-level variables** | | |
| **Child's age (months)** | | |
| 0–5 | 0.47 (0.18 - 1.22) | 0.118 |
| 6–11 | 1.56 (0.59 - 4.10) | 0.371 |
| 12–23 | 0.87 (0.47 - 1.63) | 0.671 |
| 24–59 | Ref. | |
| **Child's sex** | | |
| Male | 2.23 (1.33 - 3.74) | **0.002** |
| Female | Ref. | |
| **Child's birth size (recall)** | | |
| Large | 0.30 (0.17 - 0.54) | **<0.001** |
| Average | 0.36 (0.20 - 0.66) | **0.001** |
| Small | Ref. | |
| **Birth order (continuous)** | 1.13 (0.96 - 1.34) | 0.138 |
| **Child breastfeeding status** | | |
| Still breastfeeding | 0.13 (0.05 - 0.34) | **<0.001** |
| Ever breastfed, not currently breastfeeding | 0.19 (0.09 - 0.39) | **<0.001** |
| Never breastfed | Ref. | |
| **Maternal-level characteristics** | | |
| **Education** | | |
| No education | 5.14 (1.16 - 22.75) | **0.031** |
| Primary | 4.88 (1.36 - 17.58) | **0.015** |
| Secondary | 2.55 (0.74 - 8.80) | 0.139 |
| Higher (tertiary) | Ref. | |
| **Breastfeeding (currently)** | | |
| No | 0.66 (0.26 - 1.67) | 0.385 |
| Yes | Ref. | |
| **Mean women dietary diversity** | | |
| **Minimum Dietary Diversity-women** | | |
| Inadequate (< 5 groups) | 2.53 (1.50 - 4.26) | **<0.001** |
| Adequate (≥ 5 groups) | Ref. | |
| **Household-level characteristics** | | |
| **Sex of household head** | | |
| Male | 0.69 (0.38 - 1.26) | 0.227 |
| Female | Ref. | |
| **Type of toilet** | | |
| Improved | 0.65 (0.29 - 1.46) | 0.297 |
| Unimproved | 0.71 (0.36 - 1.42) | 0.332 |
| Open defecation | Ref. | |
| **Household wealth** | | |
| High | 0.13 (0.05 - 0.33) | **<0.001** |
| Middle | 0.66 (0.32 - 1.38) | 0.271 |
| Low | Ref. | |
| **Urbanicity** | | |
| Rural | 0.34 (0.19 - 0.61) | **<0.001** |
| Urban | Ref. | |

*(Continued)*

**Table 4.** (Continued)

| Variables | AOR | P value |
|---|---|---|
| **Climatic zones** | | |
| Savannah | 0.99 (0.42 - 2.32) | 0.973 |
| Forest | 1.26 (0.65 - 2.46) | 0.493 |
| Coastal | Ref. | |

**Model fit statistics**: Nagelkerke $R^2$ = 20.0%; Wald test =133.1, p < 0.001.

for the increasing maternal overweight/obesity in Ghana could be varied. First, it is characteristic of a country undergoing economic and nutrition transition to experience some increases in overnutrition rates akin to postulations by Popkin et al. (2012) [43]. This is mainly attributable to the occurrence of an overcorrection due to the prolonged exposure to chronic undernutrition. This tendency to overcompensate under these circumstances is also advanced by Matrins et al., 2011 [44], who reported that malnourished children are more likely to suffer overweight and overweight-related non-communicable diseases later in life due to early-life undernutrition. The second reason for the increasing maternal overweight may be an outcome of socio-cultural perceptions or misconceptions. In most parts of the developing world including Ghana, weight gain, particularly among women, is perceived as prestigious, and sometimes associated with affluence [45]. To this effect, women who gain weight are thought to be living a good life and a not-so-good life if they lose weight, irrespective of the reason for the weight loss.

Meanwhile, the reason(s) for the general decline in undernutrition, and anaemia rates in Ghana may be ascribed to, first, the building of the human resource capacity of practitioners and service providers. And second, the deliberate and concerted efforts at reducing malnutrition rates through direct programming and nutrition interventions by relevant stake-holders. The promotion of optimal infant and young child feeding practices; including exclusive breastfeeding for infants less than 6 months, timely and appropriate complementary feeding for children and lactation support mothers, are examples of such interventions [46]. Other interventions include Iron and folic acid supplementation aimed at reducing anaemia among pregnant and lactating women; the growth monitoring and promotion meant at tracking child growth and providing nutrition support when needed; the high dose vitamin A supplementation for women and children; the flour and vegetable oil fortification for general populace to increase micronutrient intake; the nutritional support for vulnerable groups such HIV and TB clients; and the Community-based Management of Acute Malnutrition (CMAM) [47–49] among others. The CMAM for instance, provides as a comprehensive undernutrition management strategy that starts from case tracing at the community level through to in-patient care at Government hospitals. These interventions have been running for a significant part of the past two decades with some being in existence for well over 3 decades [47]. Their co-ordination and implementation are governed by the Ghana Nutrition Policy, and the Ghana Public Health Act 2012 [47,50]. It should however be noted that despite the general reductions in undernutrition indicators, their current prevalence remains above internationally acceptable 'low thresholds' for stunting, wasting, underweight and anaemia which are currently pegged at 10%, 5%, 7.5% and 20%, respectively [5,51].

At the household, maternal and child levels, this study found several factors that were associated with the pooled TBM among mother-child pairs in Ghana. One such factor is infant breastfeeding. This study revealed that breastfeeding reduced the likelihood of experiencing TBM similar to a study by Chilot et al., 2023 [8]. Breast milk is comprised of the requisite nutrients and relevant immune factors that not only prevent micronutrient deficiencies but also help protect against infections that exacerbate undernutrition outcomes [52,53]. As a result, breastfed infants are less likely to be stunted, wasted or suffer micronutrient deficiencies, particularly in low-income conditions [54]. It has been established that breast-feeding also provides some protection against childhood obesity, because breastfed children are less likely to consume excessive calories [55]. Consistent with previous studies male children and children with low birth sizes had a higher

likelihood of experiencing TBM [9,29,30]. The higher nutrient requirements of male children relative to their female counterparts make male children more susceptible to undernutrition-related challenges [56]. Also, a disadvantaged early start occasioned by poor nutrient reserves associated with low-birth-size children could mean such children are more vulnerable to micronutrient deficiencies, undernutrition and later life obesity – the components of TBM.

Nutrition education that promotes the consumption of diversified food is a better positive predictor of nutritional status than food quantity. Yet most nutrition interventions, particularly in low-resourced regions, still focus on providing enough calories rather than meeting micronutrient requirements [57]. It is therefore not surprising, given the undernutrition rates, that women with no education and inadequate dietary diversity were independently more likely to suffer TBM. The observed associations could be attributed to the concept espoused by some studies that higher educational levels more often translates into greater nutritional knowledge, that may contribute to healthier food choices and better nutrition outcomes for both mother and child [58,59].

The other associated factors include wealth and urbanicity where high wealth households and rural households were less likely to suffer TBM. This instance brings to the fore the notion that the urban poor may be disproportionately affected by TBM in the wake of nutrition transition and urbanization. A notion that is promulgated by Popkin et al. In their seminal work on the global nutrition transition [43], they argue that urbanization is directly linked to dietary transition where locally available traditional diets rich in fibres, whole grains and green vegetables are replaced with highly processed diets when the rural dwellers move to urban cities. This shift, which occurs predominantly among urban poor, exposes the urban poor to high-calorie diets and also deprives them of adequate micronutrients. Their exposure to cheap, processed, salt-laden, added-sugar, and high-fat foods in urban areas is further exacerbated by aggressive marketing campaigns, mass media and the lack of a proper regulatory framework [43]; making these diets more desirable and economically accessible to many low-income earners. Suffice it to say that wealth could have protective tendencies against malnutrition, in that it may facilitate access to diverse diets, and also enhance access to healthcare, and education [31]. However, increasing obesity rates among wealthier populations in some countries suggest that while high wealth may reduce undernutrition and micronutrient deficiencies, it may not fully shield against the risks of overweight and obesity due to the aforementioned nutrition-transition-related factors.

## Public health implications

This study not only bridges the literature gap on the TBM in Ghana but also has significant implications for practice and policy. For instance, given that exclusive breastfeeding in Ghana is on a marginal decline, measures targeting breastfeeding practices needs strengthening [22]. Such measures could include the increase in maternity leave to six months so mothers can have the full benefit of the entire six months to breastfeed. Another strategy could be the re-introduction of the mother-to-mother support groups at the community level, where older women would guide and encourage younger mothers on the benefits of breastfeeding. This is particularly relevant as a study in Northern Ghana, found that older multiparous mothers were more likely to breastfeed as compared to younger mothers [60]. If Ghana is to achieve the sustainable development goal 2.2 by ending all forms of malnutrition, then stakeholders should put in measures to stem the tide of rural-urban migration by providing the requisite amenities at the rural level. It should, among other livelihood empowerment programs, strengthen the Ghana livelihood empowerment against poverty program that seeks to support economically disadvantaged households with cash handouts. Relevant stakeholders across multiple sectors should also create an environment that ensures equal opportunity for all in order to enhance equitable wealth distribution. Maternal education using behaviour change communication on the need for diversified food intake should also be re-iterated at the antenatal care level as this has the tendency to improve maternal nutrition and neonatal outcomes. Also, interventions with integrated programs targeting shared drivers (e.g., food quality, inequity) may be more effective than siloed approaches addressing undernutrition, obesity, or micronutrient deficits separately. Finally, there is the need to consider repurposing existing nutrition interventions. This would ensure that supplement-based or food-based programs do not inadvertently increase maternal or child overweight or obesity.

## Limitations

The Ghana DHS datasets are comprised of variables that are measured using cross-sectional methods, as such this study is limited to descriptions of observed associations and not causal inferences. Some variables such as birth size and food consumed over the past 24 hrs, were subjectively measured using mothers' recall and as such could be subject to recall bias. However, given the very large sample size, the effects of these biases could be insignificant as it is expected that they would be uniformly distributed among both sides of the main dependent variable. Also, the use of pooled estimates from two decades of DHS datasets means that our findings can even out year-to-year fluctuations caused by transient factors such as pandemics, providing medium-to-long-term findings that are important for policy formation and intervention implementation. The DHS is a nationally representative dataset; hence our findings are generalizable to all mother-child-pairs in Ghana.

## Conclusion

This study provides important insights into the trends and determinants of TBM in Ghana, highlighting both progress and persistent challenges. The findings indicate that despite the significant reductions in undernutrition and anaemia indicators, the TBM rate in Ghana has stagnated with very marginal reductions over the past two decades. This is largely driven by the consistent increases in maternal overweight and obesity rates. Breastfeeding, being a male child, large birth size, maternal education, dietary diversity, household wealth and urbanicity are the factors associated with the TBM. Accordingly, putting in place mechanisms to promote breastfeeding, improve diversified food intake, ensure equitable distribution of resources with emphasis of rural empowerment would help provide a unifying solution to the TBM challenge. Until such a time these systemic changes are made, health education that uses behaviour change communication targeting women can achieve a great deal.

## Acknowledgments

The authors would like to acknowledge the ICF, the USAID and Ghana Statistical Service for granting us access to these datasets.

## Author contributions

**Conceptualization:** Hammond Yaw Addae, Rafatu Tahiru, Fusta Azupogo, Martin Nyaaba Adokiya.

**Data curation:** Hammond Yaw Addae, Fusta Azupogo, Martin Nyaaba Adokiya.

**Formal analysis:** Hammond Yaw Addae, Rafatu Tahiru, Afizu Alhassan, Abdul-Ganiu Fuseini, Mohammed Iddrisu, Wilhelmina Mensah, Fusta Azupogo, Martin Nyaaba Adokiya.

**Methodology:** Hammond Yaw Addae, Rafatu Tahiru, Afizu Alhassan, Abdul-Ganiu Fuseini, Mohammed Iddrisu, Wilhelmina Mensah, Fusta Azupogo, Martin Nyaaba Adokiya.

**Supervision:** Hammond Yaw Addae, Fusta Azupogo, Martin Nyaaba Adokiya.

**Visualization:** Hammond Yaw Addae, Afizu Alhassan, Abdul-Ganiu Fuseini, Mohammed Iddrisu, Wilhelmina Mensah.

**Writing – original draft:** Hammond Yaw Addae, Rafatu Tahiru, Afizu Alhassan, Mohammed Iddrisu, Wilhelmina Mensah, Fusta Azupogo, Martin Nyaaba Adokiya.

**Writing – review & editing:** Hammond Yaw Addae, Rafatu Tahiru, Afizu Alhassan, Abdul-Ganiu Fuseini, Mohammed Iddrisu, Wilhelmina Mensah, Fusta Azupogo, Martin Nyaaba Adokiya.

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
