## [Decision Letter · Decision Letter 0]

2 Jul 2025

PGPH-D-25-00980

Trends and determinants of the triple burden of malnutrition in Ghana; analyses of two decades of Demographic and Health Survey datasets

Dear Mr Addae,

Thank you for submitting your manuscript to PLOS Global Public Health. After careful consideration, we feel that it has merit but does not fully meet PLOS Global Public Health’s publication criteria as it currently stands. Therefore, we invite you to submit a revised version of the manuscript that addresses the points raised during the review process.

EDITOR: Please insert comments here and delete this placeholder text when finished. Be sure to:

Please ensure that your decision is justified on PLOS Global Public Health’s publication criteria  and not, for example, on novelty or perceived impact.==============================Please submit your revised manuscript by 22/07/2025. If you will need more time than this to complete your revisions, please reply to this message or contact the journal office at globalpubhealth@plos.org. Please include the following items when submitting your revised manuscript:

We look forward to receiving your revised manuscript.

Kind regards,

Dickson Abanimi Amugsi, PhD

Academic Editor

Journal Requirements:

Additional Editor Comments (if provided):

Reviewers' comments:

Reviewer's Responses to Questions

**Comments to the Author**

1. Does this manuscript meet PLOS Global Public Health’s publication criteria ? Is the manuscript technically sound, and do the data support the conclusions? The manuscript must describe methodologically and ethically rigorous research with conclusions that are appropriately drawn based on the data presented.

Reviewer #1: Yes

Reviewer #2: Yes

Reviewer #3: Yes

Reviewer #4: Yes

2. Has the statistical analysis been performed appropriately and rigorously?

Reviewer #1: Yes

Reviewer #2: Yes

Reviewer #3: Yes

Reviewer #4: Yes

3. Have the authors made all data underlying the findings in their manuscript fully available (please refer to the Data Availability Statement at the start of the manuscript PDF file)?

Reviewer #1: Yes

Reviewer #2: Yes

Reviewer #3: Yes

Reviewer #4: Yes

4. Is the manuscript presented in an intelligible fashion and written in standard English?

Reviewer #1: Yes

Reviewer #2: Yes

Reviewer #3: Yes

Reviewer #4: Yes

5. Review Comments to the Author

Reviewer #1: Trends and determinants of the triple burden of malnutrition in Ghana; analyses of two decades of Demographic and Health Survey datasets

Line 73: “As a result, the concurrent existence of maternal micronutrient deficiencies and obesity in an individual could burden health systems and retard progress in achieving nations’ set health and developmental targets.” Consider omitting one of the bolded words as it sounds a bit redundant.

Line 76: “The co-existence of malnutrition also has a potential to cloud the judgement of public 77 health experts as to which malnutrition type deserves urgent attention within the context of 78 scarce resources, particularly in the presence of undernutrition and overnutrition.” Consider omitting, its redundant

Line 79: “While LMICs are more likely to be disproportionately affected by the TBM due to the pronounced effects of a rapid nutrition transition, and more recently the adverse effects of climate change, there is paucity of literature on the factors that drive TBM at the household, maternal and child-level in Ghana”

Climate change, as it currently stands in the sentence, seems forced because there is no mention of macro level factors (household, maternal and child-level only), so consider omitting or rephrasing

Line 93: “As a comparatively new concept, that lacks scholarship within previous maternal and child malnutrition discourse in Ghana, it is important to ascertain the trend of TBM over the past two decades.”

First half of this sentence needs rephrasing.

Line 184: “Likewise, an underweight child and a woman who is underweight and anaemic in the same household could also be classified as a household with TBM.”

This does not satisfy the TBD, as both mom and child here are undernourished. There is no overnutrition in this scenario, so it qualifies as a double burden.

Line 206: “maternal education, religion, ethnicity”

Categories need to be mentioned, as done with the other variables.

Line 210: “This was calculated based on 24 hr-food consumption patterns from the FAO & FHI 10-food group classifications [35]; 1) grains, white roots and tubers, and plantains, 2) beans, peas and lentils, 3) nuts and seeds 4) milk and milk products 5) meat, poultry and fish, 6) eggs, 7) dark green leafy vegetables 8) other vitamin A rich fruits and vegetables, 9) other vegetables and 10) other fruits.”

For standardization purposes, since the different food groups were mentioned for women’s diet diversity score, the same needs to be done for the child’s different food groups earlier.

Line 281: “74.7% of children aged < 24 months were still breastfeeding,”

Why wasn’t exclusively breast fed further considered? Given the WHO recommendations and the critical role it plays in malnutrition.

Line 307: “Mothers overweight had a pooled prevalence of 35.2% (34.2 - 26.2).”

For standardization purposes, why hasn’t the trend for overweight mentioned as well? (up from x)

Line 407: “Where malnourished children are more likely to suffer overweight and overweight-related non-communicable diseases later in life due to early-life undernutrition”

This is not a sentence, it needs revision.

Reviewer #2: The manuscript tackles a relevant topic but lacks novelty in its main claims. Several assertions are made without proper citation, and the literature review is thin, failing to place the study adequately within existing research. The analysis is shallow, with limited interpretation of regression results and no control for key confounding variables. Lastly, the discussion includes claims that are not fully grounded in the presented data. A major revision is needed to address issues of depth, referencing, and transparency.

Reviewer #3: I congratulates all who participated in preparing the manuscript. To me the manuscript is well thought, written and it provide new experience and insight in reviewing data sets of the long existing data sets of DHS programs not only in Ghana, but also it may attract more researchers and academia in different countries to look into their existing data. You have shown a way to others

Reviewer #4: The article was were written and cogent . However , the article will benefit from some areas that were not explored i,e interrelationship between the different types of malnutrition , analysis of TBM as outcome of complex issues . The statistical analysis section will benefit if authors could explain choice of how variables were use ( form of utilization ) and the exact choice of models used .

6. PLOS authors have the option to publish the peer review history of their article (what does this mean? ). If published, this will include your full peer review and any attached files.

**Do you want your identity to be public for this peer review?** For information about this choice, including consent withdrawal, please see our Privacy Policy .

Reviewer #1: No

Reviewer #2: No

Reviewer #3: No

Reviewer #4: No

<gdiv id="ginger-floatingG-container" style="position: absolute; top: 0px; left: 0px;"><gdiv class="ginger-floatingG ginger-floatingG-closed ginger-floatingG-posdown" style="display: block; left: 651.5px; top: 158.031px; z-index: 51;"><gdiv class="ginger-floatingG-disabled-main"><gdiv class="ginger-floatingG-bar-tool-tooltip ginger-floatingG-bar-tool-tooltip-enable">Enable Ginger</gdiv></gdiv><gdiv class="ginger-floatingG-offline-main"><gdiv class="ginger-floatingG-bar-tool-tooltip">*Cannot connect to Ginger* Check your internet connection

or reload the browser</gdiv></gdiv><gdiv class="ginger-floatingG-enabled-main"><gdiv class="ginger-floatingG-bar"><gdiv class="ginger-floatingG-bar-tool ginger-floatingG-bar-tool-disable"><ga></ga><gdiv class="ginger-floatingG-bar-tool-tooltip">Disable Ginger</gdiv></gdiv><gdiv class="ginger-floatingG-bar-tool ginger-floatingG-bar-tool-rephrase ginger-floatingG-bar-tool-rephrase_small-circle"><ga class="ginger-floatingG-bar-tool-rephrase__btn" id="ginger__floatingG-bar-tool-rephrase__btn">Rephrase</ga><gdiv class="ginger-floatingG-bar-tool-tooltip ginger-floatingG-bar-tool-tooltip_rephrase">Rephrase with Ginger (Cmd+⌥+E)</gdiv></gdiv><gdiv class="ginger-floatingG-bar-tool ginger-floatingG-bar-tool-mistakes"><ga>0</ga><gdiv class="ginger-floatingG-bar-tool-tooltip">Log in to edit with Ginger</gdiv></gdiv></gdiv></gdiv><gdiv class="ginger-floatingG__loading-popup">Ginger is checking your text for mistakes...</gdiv><gdiv class="ginger-floatingG__disabling-popup " style="display: none;"><button class="ginger-floatingG__disabling-popup-button">Disable Ginger in this text field</button><button class="ginger-floatingG__disabling-popup-button">Disable Ginger on this website</button></gdiv><gdiv class="ginger-floatingG-contentPopup" style="display: none;"><gdiv class="ginger-floatingG-contentPopup-wrap-limit">600/11405 free characters checked.Go Premium to check longer texts and entire documents</gdiv></gdiv></gdiv></gdiv>

---

## [Decision Letter · Decision Letter 1]

26 Aug 2025

Trends and determinants of the triple burden of malnutrition in Ghana; analyses of two decades of Demographic and Health Survey datasets

PGPH-D-25-00980R1

Dear Hammond,

We are pleased to inform you that your manuscript 'Trends and determinants of the triple burden of malnutrition in Ghana; analyses of two decades of Demographic and Health Survey datasets' has been provisionally accepted for publication in PLOS Global Public Health.

Best regards,

Dickson Abanimi Amugsi, PhD

Academic Editor

Reviewer Comments (if any, and for reference):

Reviewer's Responses to Questions

**Comments to the Author**

1. If the authors have adequately addressed your comments raised in a previous round of review and you feel that this manuscript is now acceptable for publication, you may indicate that here to bypass the “Comments to the Author” section, enter your conflict of interest statement in the “Confidential to Editor” section, and submit your "Accept" recommendation.

Reviewer #1: All comments have been addressed

2. Does this manuscript meet PLOS Global Public Health’s publication criteria ? Is the manuscript technically sound, and do the data support the conclusions? The manuscript must describe methodologically and ethically rigorous research with conclusions that are appropriately drawn based on the data presented.

Reviewer #1: Yes

3. Has the statistical analysis been performed appropriately and rigorously?

Reviewer #1: Yes

4. Have the authors made all data underlying the findings in their manuscript fully available (please refer to the Data Availability Statement at the start of the manuscript PDF file)?

Reviewer #1: Yes

5. Is the manuscript presented in an intelligible fashion and written in standard English?

Reviewer #1: Yes

6. Review Comments to the Author

Reviewer #1: Comments have been addressed

7. PLOS authors have the option to publish the peer review history of their article (what does this mean? ). If published, this will include your full peer review and any attached files.

**Do you want your identity to be public for this peer review?** For information about this choice, including consent withdrawal, please see our Privacy Policy .

Reviewer #1: No
